# Review of Sewage Sludge as a Soil Amendment in Relation to Current International Guidelines: A Heavy Metal Perspective

Nuno Nunes [1,2], Carla Ragonezi [1,2,*], Carla S.S. Gouveia [1,2] and Miguel Â.A. Pinheiro de Carvalho [1,2,3]

1 ISOPlexis Centre Sustainable Agriculture and Food Technology, University of Madeira. Campus da Penteada, 9020-105 Funchal, Portugal; nuno.nunes@staff.uma.pt (N.N.); csgouveia@staff.uma.pt (C.S.S.G.); miguel.carvalho@staff.uma.pt (M.Â.A.P.d.C.)

2 Centre for the Research and Technology of Agro-Environmental and Biological Sciences (CITAB), University of Trás-os-Montes and Alto Douro, 5000-801 Vila Real, Portugal

3 Faculty of Life Sciences, University of Madeira, Campus da Penteada, 9020-105 Funchal, Portugal

* Correspondence: carla.ragonezi@staff.uma.pt; Tel.: +351-291-705-002

**Abstract:** Overexploitation of resources makes the reutilization of waste a focal topic of modern society, and the question of the kind of wastes that can be used is continuously raised. Sewage sludge (SS) is derived from the wastewater treatment plants, considered important underused biomass, and can be used as a biofertilizer when properly stabilized due to the high content of inorganic matter, nitrate, and phosphorus. However, a wide range of pollutants can be present in these biosolids, limiting or prohibiting their use as biofertilizer, depending on the type and origin of industrial waste and household products. Long-term applications of these biosolids could substantially increase the concentration of contaminants, causing detrimental effects on the environment and induce hyper-accumulation or phytotoxicity in the produced crops. In this work, some critical parameters for soils and SS agronomic use, such as organic matter, nitrogen, phosphorous, and potassium (NPK), and heavy metals concentration have been reviewed. Several cases of food crop production and the accumulation of heavy metals after SS application are also discussed. SS production, usage, and legislation in EU are assessed to determine the possibility of sustainable management of this bioresource. Additionally, the World Health Organization (WHO) and Food and Agriculture Organization (FAO) guidelines are addressed. The opportunity to produce bioenergy crops, employing sewage sludge to enhance degraded land, is also considered, due to energy security. Although there are numerous advantages of sewage sludge, proper screening for heavy metals in all the variants (biosolids, soil, food products) is a must. SS application requires appropriate strict guidelines with appropriate regulatory oversight to control contamination of agricultural soils.

**Keywords:** by-products; biosolids; organic compounds; circular economy; total lifecycle assessment; biomass effect





## 1. Introduction

Nowadays, water and sewage treatments plants, coupled with the high-performance technological methods, enhance the separation of the organic residue waste, efficiently producing sludges. Due to the physical-chemical processes involved in the treatment, the sludge tends to concentrate heavy metals and poorly biodegradable trace organic compounds, as well as potentially pathogenic organisms present in wastewaters. After the proper treatment, the sludge becomes a biosolid, which is envisioned as a valuable biological resource [1]. For example, chemical purification processes of drinking water include the addition of nontoxic aluminum (Al) or iron (Fe) which chelates several contaminants, settling in the bottom, forming a sludge. When previously treated, for instance, with Fe/Mn (manganese) or phosphorous (P), which already has been tested in soils rich in copper (Cu) and lead (Pb), to reduce plant bioaccumulation, could introduce several agronomic benefits into the soil [2]. Treated sludge is defined as having undergone "biological, chemical or

heat treatment, long-term storage or any other appropriate process so as significantly to reduce its fermentability and the health hazards resulting from its use" [3]. Sewage sludge (SS) or biosolids are a major by-product of the wastewater treatment process, and they can be a source of organic matter and nutrient replenishment for the regeneration of eroded soils and improvement of soil productivity potential [4].

Sewage wastewater (SW) is normally subjected to three stages (see Figure 1), namely preliminary treatment, primary, and secondary treatments [5]. The preliminary treatment is necessary to remove objects with a diameter superior to 2 cm and heavy solids, which are not included in the biosolids. The primary treatment physically removes suspended solids and scum from wastewater, which are brought together to form a primary sludge, through a two-stage process, using pre-aeration and sedimentation, additionally providing dissolved oxygen to support microorganisms in a later stage [5]. The secondary biological treatment decreases the existing biodegradable material, using microorganisms to consume dissolved and suspended organic matter, obtaining a secondary sludge, that is separated by the density difference, due to increasing number of developed microorganisms [5]. Both primary and secondary treatments produce a sludge that is mixed to obtain a biosolid [5]. The agronomic use of biosolids is widely analyzed as one of the most useful applications for several reasons, including the incorporation of organic matter, nitrogen (N), and P in the soil, improving its physicochemical, microbial, and enzymatic properties, and thus soil fertility [6]. The use of biosolids, for instance, allows the recovery of P from domestic waste streams, which could replace, to some extent, the global demand for phosphate rock, a non-renewable resource [7]. These biosolids are often stabilized through dehydration to reduce the presence of pathogens, such as *Salmonella* spp. [8], *Escherichia coli*, and *Clostridium perfringens* [9] which could cause severe health concerns. Other more technological stabilization processes, such as thermophilic anaerobic digestion, could reduce the concentration of organic compounds and heavy metals in the biosolids, derived from the multiplicity of products used in industry, housekeeping, and other daily life activities, converting this resource into a more valuable and safe usable biomass [10].

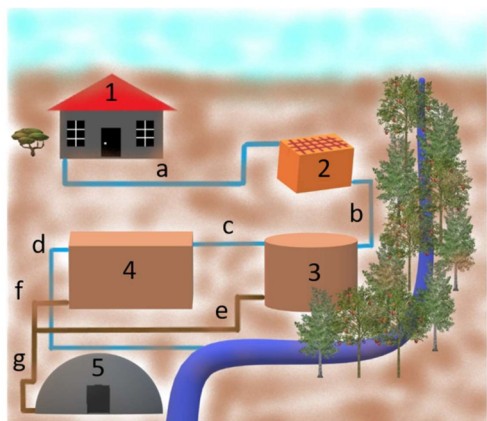

**Figure 1.** Wastewater treatment facility diagram (adapted from Demirbas et al. [5]). 1—treatment facility, 2—preliminary treatment, 3—primary treatment, 4—secondary treatment, 5—sludge dehydration, a—wastewater, b—sifted wastewater, c—wastewater without sediment, d—treated wastewater for stream discharge, e—primary sludge, f—secondary sludge, g—mixed sludges.

A wide range of contaminants can be present in these biosolids, which may limit or prohibit their use as biofertilizers, such as polynuclear aromatic hydrocarbons (PAHs), polychlorinated biphenyls (PCBs), polychlorinated naphthalene (PCN), perfluoroalkyl substances (PFASs), halogenated flame retardants (HFRs), polychlorinated n-alkanes, organochlorine pesticides, synthetic musk's, antibiotics, pharmaceuticals, and heavy metals [11–15]. Food and Agriculture Organization (FAO) guidelines [16] and the European Directive 86/278/EEC [3] describe the maximum permissible concentration of potentially toxic elements in the soil after application of SS and maximum annual rates of addition. The

previously-mentioned European directive regulates the upper values of heavy metals in soils and SS, when agricultural use is foreseen, which includes cadmium (Cd), nickel (Ni), zinc (Zn), mercury (Hg), Cu, and Pb. However, some of these values are pH dependent due to the higher mobility, resulting in a higher bioavailability to the crops. A previous systematic review which included a meta-analysis, concluded that exposure to heavy metals such as arsenic (As), Cd, Pb and Hg are positively associated with metabolic syndrome in humans, which represents an array of metabolic disorders such as hyperglycemia, high blood pressure, abdominal obesity and dyslipidemia [17]. Due to the health concerns regarding the agronomic use of SS application, which introduces heavy metals into the soils, several methods were developed to reduce this hazard. One of the early studies analyzed the use of organic acids as an extractant, such as oxalic and citric acids with two main advantages, operate at mild pH conditions (pH 3–5) and biodegradable [18]. Other processes were further developed at a laboratorial level to efficiently extract heavy metals from SS which includes supercritical fluid extraction, chemical agent treatment, plant-based washing agents, ion-exchange extraction, advanced oxidation, bioleaching, and electrokinetic processes, which are the latest to have already been tested at a pilot scale level [19]. To reduce the environmental and health risks of agricultural use of biosolids, the application of stabilization methods has been studied over the years, including composting [20], vermicomposting [21], incineration [22], adding fly ash and/or lime [23], modified clay minerals from the smectite group such as montmorillonite [24], aerobic and anaerobic digestion [10,25], and mixing with steelmaking slag [26]. Nevertheless, successive long-term applications of these biosolids could substantially increase the concentration of heavy metals, causing harmful effects to the environment, plant hyperaccumulation, and induction of phytotoxicity in the produced crops [27,28].

The application of biosolids for land reclamation and the recovery of degraded land to produce energy crops has also been exploited over the years. To maintain energy security and be prepared for the inevitable climatic changes, several plant species are tested for this purpose, among others *Populus euramericana* [29], *Cynara cardunculus* [30], *Miscanthus gigantheus*, and *Phalaris arundinacea* [31]. Research involving energy crops evaluates their calorific value and carbon content, which are indicators of its potentiality as an energy crop [32], and also determine its potential as a bio-accumulator when considering sustainable phytoremediation in contaminated land [33]. Several other reviews have been published over the years, systemizing important information that could help decide the safest and economically feasible application of sewage sludge and biosolids, evidencing the future of this biomass. Sewage sludge characterization, crop and soil effects, heavy metals accumulation, and possible risks and recommendations are often considered [34]. Other review works also include critical comparisons concerning heavy metals content between composted municipal waste and sewage sludge from several countries, metal extractability analysis, bioavailability to crops and soil microbial activity and fertility [35]. Still, more concerns related to organic contaminants and pharmaceuticals, risks to groundwater, relationship between organic matter and heavy metals availability, effect on soil salinity, eutrophication, and pathogenic organisms are also reviewed [36]. A recent work discusses some of the previous mention topics and permissible heavy metals and organic pollutants limits in composts regarding distinct countries, global production of solid wastes and biosolids, and also relates biosolids and omics study [37]. Our review complements the effort by focusing on heavy metals content, summarizing several additional reported works, concerning SS and biosolids, soil and crop analysis, discriminating individual results from distinct publications. Regarding SS and biosolids, 13 distinct cities are included and 12 for soil analysis. For food products, 18 crops, including similar species, are reviewed in their heavy metal concentration on seeds (n = 10), leaves (n = 4), and fruits (n = 3). In comparison with previous review works, we believe that discriminating individual reports and focusing only on heavy metals, provides a more thorough analysis regarding this contamination. Also, the systemization of the technical benefits and difficulties when fertilizing with SS or biosolids for crop production, evidencing current legislation and guidelines, increases

awareness on the subject. Including reports on phytoremediation and bioenergetics, we intend to complement this work, providing alternatives for food crop production when using contaminated soil or consecutive fertilization with SS or biosolids occurs, elevating soil heavy metal concentration, surpassing the current legislation upper limit.

The objectives of this work were to review the latest FAO and WHO (World Health Organization) recommendations and European legislation, concerning the production and application of biosolids for agricultural use or to improve degraded land, in order to establish sustainable management and potential use of this resource. We also review important parameters in SS, biosolids, and soil analysis, which could threaten their agricultural use, such as pH, electric conductivity, organic matter, major nutrients (N, P, potassium (K), represented as NPK), and heavy metals content. The accumulation of heavy metals in food crops when higher levels of SS or biosolids are applied. The possible use of phytoremediation strategies to overcome heavy metals toxicity effects and the production of bioenergetic crops using SS or biosolids, to improve degraded land is also discussed.

## 2. Review Methodology

This systematic review was developed by searching international databases, which included Web of Science, Science Direct, PubMed, Google Scholar, and Scopus. Two major thematic groups were selected, agricultural and biological sciences and environmental sciences. The first group included agronomy and crop science, plant nutrition, horticulture, soil science, and forestry. The second group included management, monitoring, policy and law, pollution, and waste management and disposal. The software used for bibliographic management was Mendeley for Windows (v1.19.8) for duplication search. The primary keywords selected for database search were sewage sludge, biosolids, and heavy metals. These were linked with several other words such as legislation, guidelines, amendment, food, crops, pollutants, agriculture, soil, plant, compost, analysis, health, horticulture, phytoremediation, bioenergy, contamination, and fertilization. A total of 158 research works were considered eligible. These were assessed considering the title, abstract and relevant data in order to remove irrelevant papers, followed by evaluating the quality and data presented, of which 94 research works were considered for this review. From these, 4 research papers are from before the year 2000, 29 are from 2000 to 2010, and 61 are from 2011 to 2021. Regarding the research papers with usable data for critical review, these were from 20 countries of which 6 were from the EU. Major research works used in this review were developed in India (n = 7), followed by Spain (n = 5) and China (n = 3). Two research papers were identified for each of the following countries, Brazil, Pakistan, and Sweden. Australia, Denmark, Finland, Morocco, Egypt, Poland, Tunisia, UK, USA, France, Sri Lanka, Turkey, and Algeria were represented with one report each. In this review, we also included sewage sludge, biosolids, and heavy metals current legislation and guidelines reports.

## 3. Legislation

In many countries, sewage sludge, an inevitable byproduct of municipal wastewater-treatment plant operation, is a key problem due to its increasing production and the impact associated with its disposal [38]. Regarding worldwide institutions, FAO and WHO provide their own guidelines. In the FAO's document regarding wastewater treatment and its use in agriculture, in Section 6 (Agricultural use of Sewage Sludge), point 6.2 (Sludge Application), the authors present maximum permissible concentrations of potentially toxic elements (PTE) in the soil after the application of sewage sludge [16]. This document presented the maximum permissible concentrations of potentially toxic elements in the soil after the application of sewage sludge and the maximum annual rates of addition. Among usual PTEs such as zinc and mercury, the document adds information about parameters such as molybdenum (Mo), selenium (Se), fluoride (F), and As that are not subject to the provisions of Directive 86/278/EEC [3]. Additionally, is possible to find examples of effective sludge treatment processes and further information about sewage sludge and crops.

Extensive work for WHO was prepared by Chang et al. [39] entitled 'Developing Human Health-Related Chemical Guidelines for Reclaimed Water and SS Applications in Agriculture'. At the beginning of the document the authors summarize the maximum permissible pollutant concentrations in the receiving soils for organic and inorganic compounds, although throughout the text information about local limits is provided. Limit values that did not appear in the FAO document and EU directive (see below), are presented here, such as Silver (Ag), Boron (B), Beryllium (Be), Titanium (Ti), and Vanadium (V) (see Table 1).

**Table 1.** Maximum permissible pollutant concentrations, maximum permissible concentration of potentially toxic elements and limit values for heavy metal concentrations is soil from WHO, FAO and the 86/278/EEC Directive.

| Heavy Metal Limits | WHO—World Health Organization | * FAO—Food and Agriculture Organization | # Directive 86/278/EEC |
|---|---|---|---|
| | Maximum Permissible Pollutant Concentrations in the Receiving Soils (mg kg$^{-1}$) | Maximum Permissible Concentration of Potentially Toxic Elements in Soil (mg kg$^{-1}$ Dry Solids) | Limit Values for Concentrations of Heavy Metals in Soil (mg kg$^{-1}$ of Dry Matter of Soil) |
| Arsenic (As) | 8 | 50 | - |
| Cadmium (Cd) | 4 | 3[5] | 1 to 3 |
| Chromium (Cr) | - | 400 (prov.) | - |
| Copper (Cu) | - | 80 | 50 to 140 |
| Fluoride (F) | 635 | 500 | - |
| Lead (Pb) | 84 | 300 | 50 to 300 |
| Mercury (Hg) | 7 | 1 | 1 to 1.5 |
| Molybdenum (Mo) | 0.6 | 4 | - |
| Nickel (Ni) | 107 | 50 | 30 to 75 |
| Zinc (Zn) | - | 200 | 150 to 300 |
| Silver (Ag) | 3 | - | - |
| Boron (B) | 1.7 | - | - |
| Beryllium (Be) | 0.2 | - | - |
| Barium (Ba) | 302 | - | - |
| Selenium (Se) | 6 | - | - |
| Antimony (Sb) | 36 | - | - |
| Titanium (Ti) | 0.3 | - | - |
| Vanadium (V) | 47 | - | - |

* maximum permissible concentrations of potentially toxic elements in soil after application of SS and considering pH = 5.0 < 5.5. # soil with a pH of 6 to 7.

According to the European Commission, the wastewater treatment generated almost 8 Mt of sludge in 2016, and agriculture was the main channel for wastewater sludge reutilization, absorbing around half in most EU member states [40]. The application of sewage sludge to land in member countries of the European Economic Commission (EEC) is governed by Council Directive No. 86/278/EEC (Council of the European Communities 1986) [3]. This directive seeks to encourage the use of biosolids in agriculture and regulates its utilization, preventing harmful effects on soil, vegetation, animals, and humans. The EU commission's regulation established a ban on untreated sludge and imposed specific rules for sampling and analysis of sludge and soil, to prevent harmful effects. It establishes requirements for keeping detailed records of the quantities of sludge produced, the quantities used in agriculture, the composition and properties of the sludge, the type of treatment, and the places where the sludge is used. The parameters are subject to the provisions of the Directive include the following, in mg kg$^{-1}$ of dry matter: chromium (Cr), Cd, Cu, Ni, Pb, Zn, and Hg. Additionally, analysis should cover the following parameters: organic matter, pH, N, and P.

Concerning human consumption and public health, the Commission Regulation (EC) No. 1881/2006 of 19 December 2006 [41], establishes maximum levels for certain contaminants in foodstuffs to keep them at acceptable toxicological levels.

In an EU assessment roadmap on the Evaluation of the Sewage Sludge Directive 86/278/EEC from June 2020 [40], it was stated that: "The use of sludge in agriculture is an effective alternative for chemical fertilizers, especially phosphorus fertilizers. The importance of recycling of materials, in line with circular economy principles, is highlighted as a priority area under the European Green Deal and the Circular Economy Action Plan (CEAP). However, it is important that what is used as a resource is not contaminated, otherwise, recycling will result in increased pollution of soil, water, and/or air. This is also in line with the Commission's zero pollution ambition announced in the European Green Deal (a strategy is expected in 2021)".

Figure 2 presents the number of SS produced and applied in the EU from 2008 to 2017, with a focus on the European management of the topic of biosolids [42].

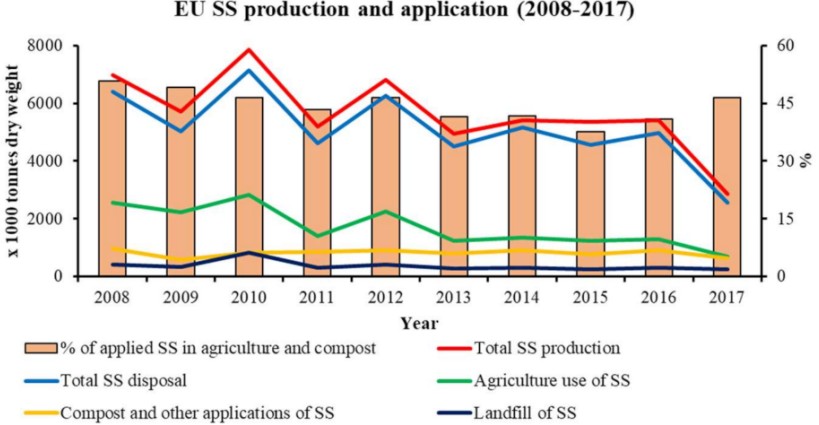

**Figure 2.** EU sewage sludge production and application from 2008 to 2017 [42].

This 10-year analysis shows that the percentage of SS applied in agriculture and compost gradually varied, being the year 2008 with the highest value (±52%) and 2015 with the lowest (±37%). The discrepancy among years could have many causes, some regarding new directives and legislation's implementation. For example, the slow decline in percentage from 2008 could be explained by the introduction of the waste hierarchy, with the Directive 2008/98/EC [43]. In 2015, the percentage started to increase again. Such events could be linked to the fact that in recent years the European Commission began to completely rethink the waste management strategy. In 2015, the EC adopted a Circular Economy Action Plan that establishes a concrete and ambitious program of action, with measures covering the whole cycle: from production and consumption to waste management and the market for secondary raw materials, and a revised legislative proposal on waste [44]. Total sewage disposal (blue line) and the total production of biosolids (red line) had variations that, in some years, followed the percentage of applied SS (2008–2012), but in some cases presented an opposite trend (2017). The application of compost and other SS (yellow line) and landfill of SS (dark blue line) were practically the same over the years. The use of SS (green line) in agriculture was surprisingly higher in the early years than in the later years. This last observation, together with the percentage of SS applied in agriculture and compost, is important to access the number of biosolids applied directly for agricultural purposes in the EU.

## 4. Sewage Sludge and Biosolids Analysis

Sewage sludge and biosolids could be considered a double-edged sword. On one hand, it represents a potential source of nutrients for plant growth and a material that can be used to improve the physical properties of the soil. However, on the other hand, it may also contain a range of potentially toxic metals, which could be dangerous for human health, primarily through the consumption of plants grown in sludge-enriched soils. Industrial effluents are a significant source of heavy metals in SS and biosolids because

the concentration of some of those elements is usually extremely high in industrial sludges. Most organic contaminants and pharmaceutical products in SS are concentrated after wastewater treatments, by particle precipitation. There is a wide variation of heavy metals limit values, even between similar geographical areas, type of amendment, and applied concentration of biosolids. The pH values, the amounts of NPK nutrients, heavy metals in SS from several studies, and different locations are presented in Table 2. These values were compared and discussed according to the Council Directive No. 86/278/EEC [3] due to the limiting regulations that control the contaminants in SS for agriculture purposes. The outliers were underlined and marked in bold.

According to the compilation of the results, most applied SS have values of pH, NPK, and heavy metals within the range stipulated in Directive No. 86/278/EEC [3]. In the case of Kong et al. [22] in Hebei (China), low pH values that were marginally lower than 6 were observed. According to article 8 of the Directive, when pH values are below 6, it is necessary to take into account the increased mobility and availability of heavy metals to the crop. The authors' main objective was to use a metabolomic strategy to evaluate the phytotoxicity of SS biochar. The results revealed that the biosolid pyrolysis into biochar altered the properties, reduced the toxicity towards wheat, and promoted the species growth.

A pot experiment was conducted using the amendment ratios for agricultural soil in India, by mixing 20% and 40% (*w/w*) of SS to study the use of SS amendment and the consequent contamination of palak (*Beta vulgaris* var. Allgreen H-1) by heavy metal for a leafy vegetable [45]. The biosolids amendment led to a significant increase in the concentrations of Pb, Cr, Cd, Cu, Zn, and Ni in the soil [45]. The concentration of Cd in soil was found to be above the Indian legal limits according to Characterization Of MSW Compost and its Application in Agriculture (Central Pollution Control Board— CUPS/59/2005-06 from 2006), and the values described in Directive No. 86/278/EEC [3] for both the amendment ratios. Limit values of the Directive range from 20 to 40 mg of Cd per kg of biosolids and in Singh and Agrawal's [45] work, this element showed values of $154.5 \pm 2.52$ mg of Cd per kg of sewage sludge.

It is important to highlight works that present values that are significantly below the limits according to the directive [3]. For example, in the case of Seleiman et al. [46] from Helsinki (Finland), they observed that the mixture of SS with peat improved the characteristics of the SS as a fertilizer. This observation was due to an increase in leaf area and accumulation of biomass of sludge–peat mixture fertilized in bioenergy crops, compared to plain sewage sludge. Still, Cd had a value of 0.4 (limits 20 to 40 mg $kg^{-1}$), Cu was 270 (1000 to 1750 mg $kg^{-1}$), and Zn was 30 (2500 to 4000 mg $kg^{-1}$) (Table 2). Moreover, the authors proposed that the heavy metal stress imposed by the pollutants on sludge can be ameliorated by mixing the sludge with the peat.

Sewage sludge improved the morphological and ecophysiological parameters of sunflower (*Helianthus annuus* L.) seedlings, as well as the soil chemical characteristics in Meknès-Saïs, Morocco, according to Mohamed et al. [6]. In this study, Hg had a value of $0.44 \pm 0.1$ (limit 16 to 25 mg $kg^{-1}$), and Pb was $81 \pm 4.5$ (limits are 750 to 1200 mg $kg^{-1}$) (Table 2).

In summary, these works evidence that pyrolysis could be applied to biosolids to reduce toxicity to the cultivated crops. Also, mixing SS with other materials (such as peat) could improve sewage sludge fertilization characteristics and still ameliorate the stress imposed by the heavy metals. Nonetheless, sewage sludge could improve the morphological and ecophysiological parameters in seedlings (such as sunflower plants) and soil chemical characteristics.

**Table 2.** Sewage sludge analysis.

| Country/City | pH | N | P | K | Cd | Cr | Cu | Hg | Ni | Pb | Zn | Ref. |
|---|---|---|---|---|---|---|---|---|---|---|---|---|
| | | g kg$^{-1}$ DW | | | mg kg$^{-1}$ DW | | | | | | | |
| Argentina/Buenos Aires | - | 2.7 ± 0.47 | 7.7 ± 1.4 | 1.4 ± 0.3 | 3.9 ± 1.9 | 155.6 ± 51.8 | 360.3 ± 80.9 | - | 85.5 ± 76.1 | 322.7± 151.6 | 1526 ± 523 | [47] |
| Australia/Brisbane | - | - | - | - | 1.8 | 16.7 ± 0.3 | 447.7 ± 1.5 | - | 19.8 ± 0.2 | 41.7 ± 0.3 | 830.6 ± 4.2 | [48] |
| Brazil/Jaboticabal | - | 34.08 | 21.62 | 1.9 | 11 | 808 | 722 | - | 231 | 186 | 2159 | [49] |
| China/Hebei | **5.8** | 56.6 | 9.8 | - | 0.3 | 83.51 | 221.9 | - | 32.62 | 73.31 | | [22] |
| China/Taiyuan | - | - | - | - | 22.1 | 245.8 | 1122 | 20.6 * | - | 118.5 | 3059 | [50] |
| Denmark/Copenhagen | 7.7 | 47 | 33 | - | 1.4 | 98 | 244 | - | 31 | 178 | 1041 | [51] |
| Finland/Helsinki | 7.2 | 0.031 | 0.026 | 0.002 | 0.4 | 30 | 270 | - | 20 | 20 | 470 | [46] |
| India/New Delhi | 6.4 | 18 | 16.1 | 1.83 | - | - | 173 | - | - | 78 | 1853 | [52] |
| India/Uttarakhand | 9.0 | - | 0.216 ± 0.002[#] | - | 10.24 ± 0.14 | 8.63 ± 1.06 | 18.96 ± 1.09 | - | - | 9.33 ± 1.01 | 11.25 ± 1.00 | [53] |
| India/Varanasi | 7.0 | 17.3 ± 0.2 | 0.717 ± 0.06 | 0.209 ± 0.002 | **154.5 ± 2.52** | 35.5 ± 0.76 | 317.7 ± 1.92 | - | 18.9 ± 0.09 | 60 ± 5.77 | 785.3 ± 16.69 | [45] |
| Morocco/Meknès-Saïs | 6.1 | 52.2 ± 2 | 0.586 ± 0.018 [#] | 0.920 ± 0.021 | 1.15 ± 0.2 | 32.8 ± 2.4 | 17.9 ± 1.2 | 0.44 ± 0.1 | 20.9 ± 1.7 | 81 ± 4.5 | 215 ± 12.4 | [6] |
| Pakistan/Multan | 6.9 | 14.6 | 13.38 | - | 5.5 | - | 145 | - | 35 | 20 | - | [54] |
| Pakistan/Multan | 7.6 | 6 | 13 | 13 | 26 | - | - | - | 160 | 13 | - | [55] |
| Spain/Alicante | 6.5 | 2.48 | 5.62 | 7.89 | 1.6 | 16.6 | 157 | n.d. | n.d. | 40.8 | 470 | [20] |

Underlined and in bold are the values that exceed the maximum allowable concentrations for heavy metals according to the Council Directive 86/278/EEC of June 12, 1986 [3]. n.d.—not determined. *—ppb (parts per billion). [#]—Extractable phosphorus—Olsen Method. Ref.—References.

## 5. Soil Analyses

Due to the composition of SS, the application into the soil may be the least energy-consuming and the most cost-effective means of disposing of or using sludge to improve soil fertility. This type of application, which is in line with the concept of a circular economy, considers the following hierarchy for waste management: prevention, reuse, recycling, recovery, and disposal of waste.

However, the positive effects (e.g., yield) and negative aspects (e.g., high concentrations of heavy metals) must be accounted for when SS or biosolids are recycled to the soil as an amendment. The challenge is to find that ideal concentration of SS, where it is possible to retrieve the best results but without soil deterioration.

As for the analysis of soils, before SS application, there is a wide variation in the heavy metals limit values, even from similar geographical areas. The pH values, electrical conductivity (EC), amount of organic matter, NPK, and heavy metals present in soils, evaluated in different locations, are shown in Table 3. The values of heavy metals that exceed the maximum allowed, described in the Council Directive No. 86/278/EEC [3] for agricultural soils, were underlined and marked in bold. According to this compilation of results, most articles described soil values of pH, NPK, and heavy metals within the range stipulated by Directive No. 86/278/EEC [3].

**Table 3.** Soil analysis.

| | pH | EC | Organic Matter | N | P | K | Cd | Cr | Cu | Hg | Ni | Pb | Zn | Ref. |
|---|---|---|---|---|---|---|---|---|---|---|---|---|---|---|
| **Country/City** | | dS m$^{-1}$ | | g kg$^{-1}$ DW | | | | | | mg kg$^{-1}$ DW | | | | |
| China/Taiyuan | 7.2 | - | - | 1.2 | - | - | **38** | 112.5 | 34.6 | - | - | **87.7** | 81.2 | [50] |
| Egypt/Sohag | 7.9 | 0.61 | 2 | 0.053 | 0.003 | 0.077 | 0.5 | - | 11 | - | - | 7 | 32 | [56] |
| India/New Delhi | 8.4 | 0.19 | - | 0.0116 | 2.65 | 141.5 | - | - | 2.07 | - | - | 0.056 | 3.76 | [52] |
| India/Uttarakhand | 7.4 | 2.63 | - | - | - | - | 0.09 | 0.18 | 2.42 | - | - | 0.12 | 0.88 | [53] |
| India/Varanasi | 8.2 | 0.24 | - | 1.8 | 0.054 | - | 1.51 | 0.34 | 3.51 | - | 4.95 | 2.83 | 2.11 | [45] |
| Morocco/Meknès-Saïs | 8.2 | 0.1 | 9.8 | 0.7 | - | 0.271 | 0.22 | 57.5 | 1.6 | <0.1 | 21.5 | 16.2 | 3.1 | [6] |
| Poland/Silesia | 7.9 | - | - | 0.481 | 0.015 | - | 2.3 | 26.32 | 29.44 | - | 28.99 | 46.57 | 112 | [31] |
| Spain/Alicante | 7.9 | 1.64 | 1.29 | 0.0125 | 0.007 | 0.3 | 0.15 | 14.7 | 0.94 | n.d. | n.d. | 0.21 | 0.59 | [20] |
| Spain/Murcia | 8.5 | 0.16 | 6.71 | 0.66 | 0.32 | 4.76 | 0.1 | - | 7.3 | - | 14.3 | - | 24.3 | [57] |
| Spain/Santiago de Compostela | **4.8** | - | - | 5 | - | - | - | 30 | 12 | - | 20 | 36 | 78 | [58] |
| Tunisia/Tunes | 8.0 | 0.263 | - | 1.1 | - | - | - | - | 32 | - | 50 | 22 | 70 | [59] |
| U.K./Reading | 5.6 | - | 77 | - | 1.208 | - | 1.2 | 35 | 38 | 0.2 | 12 | 34 | 71 | [60] |

Underlined and in bold are the values that exceed the maximum allowable concentrations for heavy metals according to the Council Directive 86/278/EEC of June 12, 1986 [3], or/and for FAO—wastewater treatment and use in agriculture [16], or/and WHO human health-related chemical guidelines for reclaimed waster and SS applications in agriculture [39]. n.d.—not determined. Ref.—References.

Regarding the document of wastewater treatment and use in agriculture from the Food and Agriculture Organization of the United Nation (FAO), in Section 6 (Agricultural use of Sewage Sludge), point 6.2 (Sludge Application), the authors present maximum permissible concentrations of potentially toxic elements (PTE) in the soil after application of SS [16]. When considering the pH of 5.0 < 5.5, values of Cd are set in 3, Zi in 200, and Pb in 300 mg kg$^{-1}$ dry solids, for example. Depending on the element, this value can change with the increasement of pH (e.g., Zi is set on 200 for pH 5.0 < 5.5, 250 for pH of 5.5 < 6.0, and up to 450 for pH > 7.0). Parameters such as Mo, Se, F, and As, are not subject to the provisions of Directive 86/278/EEC [3].

Sewage sludge can be used for soil amendment at the rate of 40% by maintaining the soil health and maximum yield of French bean (*Phaseolus vulgaris* L.) in Uttarakhand, India [53]. The initial soil values presented in Table 2 for this work were 0.09 mg for Cd (limits are 1 to 3 mg kg$^{-1}$), 0.12 mg for Pb (limits are 50 to 300 mg kg$^{-1}$), and 0.88 mg for Zn (limits are 150 to 200 mg kg$^{-1}$). Values were also below permissible limits for FAO and WHO limits. In Taiyuan (China), Cd values higher than the Directive rules for rapeseed germination and plumelet development, barley, and Chinese cabbage yields [50] were observed. Posterior application of composted SS did not affect the rapeseed germination, and was beneficial for the plumelet development at lower application rates (<150 ton ha$^{-1}$). Also, it generated positive yield responses for barley and Chinese cabbage. However, SS

increased the concentrations of Cu and Zn in 0–20 cm of soil and had little effect on those metals in the deeper soil (>20 cm). The limit values in Directive No. 86/278/EEC [3] range from 1 to 3 mg, and the soil values presented in this work were 38 mg kg$^{-1}$. This also corroborates the limits for Cd in the Chinese legislation (20 mg kg$^{-1}$ DW for alkaline or neutral soils), which holds as the standard regulation related to sludge, the Control Standards of Pollutants in Sludges from Agricultural Use (GB 4284-84), that was promulgated in 1984 but never been amended. Further, it was also high for FAO (3 mg kg$^{-1}$ dry solids) [16] and WHO limits [39] (4 mg kg$^{-1}$). Also, in this work, Pb was slightly higher (87.7 mg kg$^{-1}$ DW) than the permissible limit only for WHO parameters (84 mg kg$^{-1}$), but not higher than the permissible limits in the Chinese legislation (GB 4284-84). Kidd et al. [56] also analyzed the soil from Santiago de Compostela, Spain, in which some values were borderline according to the Directive [3] (Table 3), namely Ni and Pb. Compared with FAO [16] and WHO [39] guidelines, the values were below the permissible limits for all analyzed heavy metals. Additionally, the pH presented a lower value (4.8) than the limits according to the Council Directive No. 86/278/EEC [3] (from 6 to 7), which could increase the bioavailability of the heavy metals to the produced crops. Even despite existing global guidelines (e.g., FAO and WHO), that can indicate the limit values for most heavy metals, and even existing continental information, such as EU directives, the importance of local decree is of the essence. Among different motives, is possible to highlight the materials that compose the SS, the treatments to neutralize possible harmful agents and type of application and purpose of use. Additionally, updating and amending the already existing directives are essential to boost the SS utilization, dimmish social prejudice, and to meet the goals related to the adequate standardization of this residue and application according to the present day.

## 6. Food Crops

The agricultural use of SS and biosolids is a worldwide effective sludge disposal technique. However, its application in agricultural soils must be well thought out, regarding human food consumption. Soil amendments with these resources have been reportedly useful in increasing the number of agro-morphological attributes and yields in different crop species [61], but it is common knowledge that this kind of biomass often contains heavy metals and toxic organic residues. Its indiscriminate use can be detrimental to soil productivity and cause harm to the food chain [37]. Further, dietary intake of heavy metal-contaminated plant food can affect human health in the long term by damaging the nervous, pulmonary, and renal systems [62]. There is a wide variation in limit values for heavy metals, even between similar geographical areas, and depends on the food crops species and the kind of amendment and applied ratios. The extent of heavy metal uptake by plants grown in soil amended with sludge/compost has been evaluated in different locations, as indicated in Table 4. The Codex Alimentarius Commission developed in a joint FAO and WHO [63] effort together with the European Union Commission Regulation No. 1881/2006 of December 19, 2006 [41] which regulates the maximum levels of contaminants in foods. Unfortunately, these could not be compared with the values in Table 4, due to the discrepancy between fresh and dry weight values present in the current legislation and revised works, respectively, without the water content information.

Kumar and Chopra [53], analyzed the presence of heavy metals in French bean (*Phaseolus vulgaris* L.) grains, cultivated in 100% SS. Also, Singh and Agrawal [64] studied the accumulation of heavy metals in mung bean (*Vigna radiata* L.) seeds, when these were cultivated in a 120 ton ha$^{-1}$ SS enriched soil in Varanasi, India. Moreno et al. [58] conducted an experiment with lettuce (*Lactuca sativa* L.) in Murcia, Spain, and observed that Cd also exceeds its maximum allowable concentration when 80 ton ha$^{-1}$ of composted SS is applied to the soil. The authors concluded that care should be exercised when sludges contaminated with Cd and Zn are used for agricultural purposes since these metals are easily absorbed by plants and exert an adverse effect on yield. The risk of heavy metal uptake through the roots and its potential risk for human health is high in horticultural crops because they are characterized by a fast growth rate when compared to other plant

species, for example, woody species [65]. Singh and Agrawal [66] ascertained that okra (*Abelmoschus esculentus* L.) presented concentrations of Cd, Pb, and Ni in fruits above the permissible limits of Indian standards, 65 days after sowing when 40% (*w/w*) of SS is applied. Latare et al. [67] observed that there was a significant increase in heavy metals content in rice and wheat with the increase of sludge application in a rice-wheat system (*Oryza sativa—Triticum aestivum*). The Cd content in rice grain was above the Indian safe limit when sludge application reached 20 ton ha$^{-1}$ or higher levels, and for Cr, Ni, and Pb at 40 ton ha$^{-1}$. In contrast, only Cd shows higher values for wheat seeds. In another study, the wheat seed species *Triticum aestivum* L. presented higher values for Pb when 75% (*w/w*) of SS was applied [57]. These authors observed that the accumulation magnitude of the elements in the grains was generally less than that in the wheat shoots and roots.

Food crop species, varieties, and cultivars differ in their physiological behavior when they are produced in soils with a high content of heavy metals [68]. Zhu et al. [69] tested 24 different cultivars of Asparagus bean (*Vigna unguiculata* subsp. *sesquipedalis* L.) and determined a significant variation in Cd accumulation in roots, stems, leaves, and fruits. Over the years, several strategies were described for plants regarding the level of heavy metal toxicity. Heavy metals present in soil are absorbed by the plant root system, together with water and minerals [70]. Krzesłowska [71] reviewed the physiological response of the plant cell wall to trace metals and described that it has an active adaptive structure under toxic metal stress, increasing its production of low-methyl esterified pectins, capable to bind with divalent and trivalent metal ions. This capability is an addition to its function as a sink for toxic trace metal, reducing its amount incorporated into the plant cell protoplast. Furthermore, Probst et al. [72] described the thickening of the root cells and some modifications in the leaves, such as swollen thylakoids with grana quantity decreased and the increase of plastoglobuli in chloroplasts when studying the faba bean (*Vicia faba* L.) response to metal toxicity. However, several strategies are adopted by plants to respond to metal toxicity. The avoidance is the primary plant effort to prevent or reduce phytotoxicity when cultivated in soils with toxic amounts of heavy metals. These toxic elements, to some extent, are accumulated in the plant cells. However, they are accumulated in different concentrations within the plant structures, as described by Zhou et al. [68]. The authors analyzed the heavy metal accumulation in 22 vegetable species to determine the human health risk of vegetable intake and verified significant concentration differences in diverse species or plant parts. Zhuang et al. [73] determined a significantly higher concentration of heavy metals in leafy vegetables such as cabbage (*Brassica oleracea* L., *Brassica chinensis* L.), and lettuce (*Lactuca sativa* L.), which accumulation decreases, according to series Cd > Zn > Cu > Pb. They also observed a decrease of Cu, Zn, Pb, and Cd average concentration from stalk > husk > grain. Additionally, Pb concentration tends to be higher in root vegetables like sweet potato (*Ipomoea batatas* L.) and taro (*Colocasia esculenta* L.).

The translocation of heavy metals into the food chain accounts for 90% of the human contact, with the remaining 10% being the result of inhalation and dermal exposure [74]. In Bangladesh, excessive heavy metal contamination in vegetables was considered to have carcinogenic risk, except for Cd and Pb [75]. Endemic upper gastrointestinal cancer, present in the Van region of Eastern Turkey, can also be correlated with a high heavy metal concentration, measured in soils, fruit, and vegetables [76]. The consumption of heavy metal-contaminated food crops is also known to compromise renal function due to the ability of kidneys to reabsorb and concentrate divalent metals [77]. This relationship was assessed through heavy metal exposure, renal biomarkers, and oxidative stress, analyzing the urine samples of 944 lactating mothers (17–48 years) and their infants (2.5–12.5 months), living in Riyadh, Saudi Arabia [78]. Their findings suggest that continuous exposure to heavy metals by the overall population could cause renal damage, which children are more prone to due to their organ's immaturity. In a systematic review of cardiovascular disease, Navas-Acien et al. [79] determined that there is enough data to conclude that Pb exposure could cause hypertension. Additionally, Weisskopf et al. [80] studied the bone Pb concentration of 330 patients and suggested that cumulative exposure to Pb could also be

linked to the risk of developing Parkinson's disease. However, the disposal of biosolids into arable land for food crop production seems to present a different behavior regarding heavy metal availability. Mossa et al. [81] analyzed the soil metal dynamics when the long-term application of biosolids occurs (100 years) in the East Midlands of England and determined that the organic matter and phosphate from the biosolids accumulated in the soil reduces the reactivity of metals. An inverse relationship was observed between the concentration of soil phosphorus and the Cd and Pb, directly correlated with Ni, Zn, and Cu, revealing a complex interaction between the intricate matrix composition of the biosolids. Previously, Hosseini Koupaie and Eskicioglu [82] also performed a health risk assessment of food crops, fertilized with biosolids application of 5 to 100 ton ha$^{-1}$, for 8 years, in one time or long-term application. A probabilistic-based analysis determined that even when Cd (for vegetables) and Cu (for rice) could contribute to the health risk, the results of hazard index (HI) indicate that no potential risk to human health is verified even when considering long-term biosolids fertilization at 100 ton ha$^{-1}$. In summary, SS used for agriculture purposes with higher Cd and Zn concentrations could exert an adverse effect on yield since these heavy metals are easily absorbed by plants. Also, horticulture crops, due to their faster growth, accumulate heavy metals faster and a continuous SS application increases the heavy metals in soils. Furthermore, different plant species, cultivars, and organs of the same plant absorb dissimilar amounts of heavy metals. A multitude of strategies is developed by plants to cope with toxic levels of heavy metals in soils. Still, although most SS applications in agricultural soils are within safe limits, the ingestion of a high concentration of heavy metals, present in food crops, could cause health concerns, which primarily affects renal function. The reactivity of the heavy metals could be significantly reduced when SS is processed into biosolids, before its agricultural application, due to the organic matter and phosphate concentration.

Table 4. Quantity of heavy metals in food crops, sewage sludge/compost application, and the location of the research.

| Specie/Plant Organ * | Cd | Cr | Cu | Ni | Pb | Zn | Sewage Sludge/Compost Application | City/Country | Ref. |
|---|---|---|---|---|---|---|---|---|---|
| | | | **g kg$^{-1}$ DW** | | | | | | |
| *Hordeum vulgare*/Seeds | - | - | 5.4 ± 0.5 | 0.060 ± 0.010 | - | 20 ± 3 | 22.3 ton ha$^{-1}$ year$^{-1}$ | Girona/Spain | [1] |
| *Hordeum vulgare*/Seeds | - | - | 2.8 | - | - | 15.3 | 150 ton ha$^{-1}$ compost (2:1, wood waste:SS) | Taiyuan/China | [50] |
| *Hordeum vulgare*/Seeds | 0.038 | - | 11.2 | 1.29 | - | 61.7 | 80 ton ha$^{-1}$ | Murcia/Spain | [83] |
| *Phaseolus vulgaris*/Fruit | **2a** | 0.4a | 0.8a | - | 0.02a | 5a | 100% | Haridwar/India | [53] |
| *Vigna radiata*/Fruit | **1.62 ± 0.13** | 1.47 ± 0.15 | 2.22 ± 0.22 | 5.67 ± 0.51 | **3.47 ± 0.35** | 22.07 ± 1.08 | 120 ton ha$^{-1}$ | Varanasi/India | [64] |
| *Brassica rapa*/Leaves | - | - | 21.9 | - | - | 23.8 | 150 ton ha$^{-1}$ compost (2:1, wood waste:SS) | Taiyuan/China | [50] |
| *Lactuca sativa*/ Leaves | **1.2** | - | 12.3 | 28 | - | 202 | 80 ton ha$^{-1}$ | Murcia/Spain | [58] |
| *Zea mays*/Seeds | - | - | 1.8 ± 0.6 | 280 ± 50 | - | 17.5 ± 0.8 | 25.4 ton ha$^{-1}$ year$^{-1}$ | Girona/Spain | [1] |
| *Zea mays*/Seeds | 0.06 | 10.55 | 5.6 | 0.98 | - | 85.8 | Pots with 2.66 kg of SS on top + 13.33 kg of Soil | Helsinki/Finland | [46] |
| *Zea mays*/Seeds | - | - | - | 2.65 | - | - | 67.5 ton ha$^{-1}$ | Jaboticabal/Brazil | [49] |
| *Brassica oleracea* cv. Blue Vantage/Leaves | 0 | 0.04 | 2 | 0.36 | 0.06 | 7.5 | 2.69 ton ha$^{-1}$ | Kentucky/USA | [84] |
| *Brassica oleracea* cv. Packman/heads | 0 | 0.05 | 3 | 0.475 | 0.1 | 18 | 0.67 ton ha$^{-1}$ | Kentucky/USA | [84] |
| *Abelmoschus esculentus*/Fruit | **21** | 1.1 | 9.4 | 7.3 | **4.3** | 34.3 | 40% (w/w) | Varanasi/India | [66] |
| *Brassica napus*/Seeds | 0.05 | 0.12 | 2.5 | 0.22 | - | 19.5 | Pots with 2.66 kg of SS on top + 13.33 kg of Soil | Helsinki/Finland | [46] |
| *Oryza sativa*/Seeds | **2.28 ± 0.03** | 6.37 ± 0.13 | - | 11.68 ± 0.49 | **0.85 ± 0.08** | 22.59 ± 2.73 | 40 ton ha$^{-1}$ | Varanasi/India | [67] |
| *Beta vulgaris*/Leaves | 25a | 3a | 22a | 5a | **2a** | 75a | 40% (w/w) | Varanasi/India | [45] |
| *Triticum aestivum*/Seeds | **1.09 ± 0.02** | 0.49 ± 0.03 | - | 2.32 ± 0.23 | n.d. | 52.44 ± 0.84 | 40 ton ha$^{-1}$ | Varanasi/India | [67] |
| *Triticum vulgare*/Seeds | 0.15 ± 0.05 | - | 8.333 ± 0.775 | - | **4.800 ± 0.087** | 16.202 ± 0.368 | 75% (w/w) | Sohag/Egypt | [57] |

a - Extrapolated from graphics. Underlined and in bold are the values that exceed the maximum allowable concentrations for heavy metals for the European Union Commission Regulation No. 1881/2006 of December 19, 2006 [41] setting maximum levels for certain contaminants in foodstuffs. * refers to the edible parts of the plants, e.g., for *Triticum aestivum* would be the seeds. ton ha$^{-1}$ year$^{-1}$—Tons per hectare per year. n.d.—not determined. Ref.—References.

### 7. Phytoremediation and Bioenergetics

For over three hundred years, phytoremediation has been a cost-effective plant-based technology for carbon sequestration and clean metal-contaminated ecosystems. Byproducts of industrial and other man-made activities incorporate heavy metals and metalloids in the environment, such as Pb, Cd, Zn, Cu, Co, As, Ni, Fe, Cr, and Se [33,85,86]. The phytoremediation strategies depend on the nature and concentration of the metal-contaminant, ecosystem physicochemical properties, and plants [87]. The model of plants to be used as ideal phytoextractors must have the ability to accumulate a range of heavy metals in their harvestable parts and also to have high biomass and fast growth rates for bioenergy purposes [88].

Currently, the use of bioenergy perennial non-edible crops, such as trees, grasses, and herbs, has been considered a sustainable approach for phytoremediation with economic returns [87]. The application of SS and biosolids could support the phytoremediation process of metal-contaminated soils, with the breakdown and leaching of organic matter leading to an increase in the plant biomass phytoremediators and improvement of soil microorganisms [89]. An eco-friendly bioenergy option for a source of renewable energy source in agricultural, industrial, transportation, and household sectors, is the remediation of contaminated lands with energy crops [33].

The selection of perennial non-edible crops for phytoremediation and biomass production has to consider their specificity, transport, accumulation, and tolerance mechanisms for different metals. Plants use up to five types of phytoremediation strategies to cope with metalliferous environments [87,90]. They can use the phytoextraction (or phytoaccumulation) as a high metal translocation rate between plant roots and upper organs, with high contaminant concentration factor in aboveground organs [91]. Phytostabilization is the use of vegetation to restrict the metal-bioavailability in soil and water, aiming for the stabilization of the metals (Cr, Cd, As, Hg, Cu, and Zn) [88]. Phytodegradation (or phytotransformation) is a decontamination process that uses enzymes in the rhizosphere or even in the plant root system, to breakdown the contaminants into less-toxic products [87]. The rhizofiltration (or phytofiltration) is used by aquatic or land plants to decontaminate aquatic ecosystems. It is a method with a low chance of atmospheric contamination because of the concentration of the metal-contaminant concentration in the roots until the saturation point, without any translocation to the shoots [88]. Finally, the phytovolatilization is a form of plants to extract pollutants (such as Hg, Se) from the soil, water, and sediments. The contaminants are absorbed by roots, converted into less-toxic compounds, and subsequently released into the atmosphere through leaf evapotranspiration, which could disturb the natural equilibrium of the atmosphere [92].

However, plants that naturally grow in metalliferous environments are considered excluders or hyperaccumulators. Excluders repel metal ions present in the soil as a survival strategy or, if the uptake occurs, the toxic effect is restrained and detoxified in the roots, with the aboveground organs being less affected. Meanwhile, the hyperaccumulators can accumulate up to a 1000-fold higher level of heavy metals in their tissues than for non-hyperaccumulating species under the same metal-contaminated matrix conditions [93]. The hyperaccumulation process starts with the metal-uptake detoxification in the root system through a metallocomplex formation, using ZIP proteins. It allows a further translocation via the apoplast or symplast, from plant underground to aboveground organs, through the sequestration, distribution, and accumulation of metals at less metabolically active cells inside the plant tissues [87,93,94]. For example, the alpine penny grass (*Thlaspi caerulescens* J. & C. Presl.) is a Cd-hyperaccumulator, known for its great capability and efficiency in phytoremediation. The root membranes have a specific Cd transporter that improves Cd root uptake and leads to high root proliferation, when in the presence of a high Cd level. The use of alpine penny grass assists the phytoextraction of Cd in industrial soils and agricultural fields. With SS added, an increment of biomass production and the metal extraction capacity is observed, with a higher shoot area that improves the metal retention capacity [92]. Their successive growth and harvest cycles can lead to a positive

outcome on reducing the risk of contamination of the food chain by Cd by lowering the transference of this toxic metal from the soil to another following edible plant crop, like lettuce (*Lactuca sativa* L.) [91]. The reed canary grass (*Phalaris arundinacea* L.) is another fascinating species for phytostabilization of metal-contaminated soil, with roots having a higher accumulation of Co and Cd and leaves with Zn and Pb. The reed canary grass was also indicated for Co and Zn environmental biomonitoring (e.g., agricultural fields, cities with high traffic contamination), with a direct and significant correlation of heavy metals organ accumulation with the levels of these elements in the environment [88].

Water hyacinth (*Eichhornia crassipes* (Mart.) Solms) and other floating aquatic weed plants, such as water lettuce (*Pistia stratiotes* L.) and duckweed (*Lemna minor* L.), are noticeable bioremediators of water polluted by heavy metals [95]. The water hyacinth has a good potential for Cd remediation, showing a phytotoxic tolerance to Cd up to 15 mg $l^{-1}$, accumulating it efficiently in the aboveground tissues [96,97]. This effective phytoremediation process in water hyacinth absorption of Cd from contaminated water could be due to a mutualistic symbiosis between the roots and arbuscular mycorrhizal fungi [98].

Both processes of contaminated ecosystems phytoremediation and the use of biosolids/compost as a waste management strategy for soil fertilization generate significant biomass for industrial purposes [87]. For example, Çelebi et al. [99] tested hydroponically the growth of native grass known as sunburst switchgrass (*Panicum virgatum* L.) in four distinct Pb-treated solutions and observed good Pb accumulation with no growth delay, considering it a good phytoremediator and fuel producer. Willow (*Salix viminalis* L.) is a phytoextractor, known as a fast-growing deep-rooting woody species, with high transport of these heavy metals to shoots, specifically for Cd, Zn, and Cu. Willow can remove up to 61 g of Cd $ha^{-1}$ $year^{-1}$ [90]. The stems from Willow clones are relevant for bioenergy production in the Swedish region, but as a counterpart, the ashes contain high concentration in Cd. To be used as a field fertilizer, the ashes undergo a purification step by removing Cd from the ash during the combustion process [90,100]. The Cd and Zn are preferably accumulated in the Willow leaves, allowing the removal of soil heavy metals with the continuous stem harvest [100]. Giant reed (*Arundo donax* L.) has shown an extensive and profound root system and high-quality biomass production in both polluted water and soil environments, with culms presenting high cellulose content, commonly used for the production of fiber, paper, biopolymers, and energy (second-generation ethanol, and biodiesel). Giant reed plants have proper support for the phytoextraction process by showing high translocation factor for Cd, Co, Pb, Fe, and Ni, with the last being the metal with the most concentration in the leaves. Additionally, it has a lower translocation factor for Zn and Cr, with a higher concentration in the root organ [92]. *Eucalyptus* spp. trees are also relevant in wood biomass production to generate firewood, boards, charcoal, pulp, and paper [101]. With the application of SS, *Eucalyptus grandis* W. Hill registered an increment of 86% in biomass content, equivalent to approximately 80 ton $ha^{-1}$, and improved the *Eucalyptus camaldulensis* Dehnh development, with no negative effects on trees' health [101,102]. Likewise, the application of biosolids compost in Cardoon (*Cynara cardunculus* L.) led to an increase in energy yield, enhancing the oil production for biofuel for the Mediterranean region [30]. However, the biomass harvested from metal-contaminated sites for industrial application needs an ecotoxicological risk assessment before use. The utmost ecologically safe solution is to use the energy production for combustion or pyro-gasification, followed by metal recovery from the fly ash, using hydrometallurgical paths [87,92]. Also, the contaminated unharvested material should be phytostabilized by covering the remediated area with hardy vegetation, to counteract the transference and metal accumulation in the topsoil through leaf decomposition [103].

## 8. Conclusions

The challenge of this review was to consolidate such a broad topic, as sewage sludge, having as basis the legislation and composition analysis of the biosolids, with further

application in soil amendment, crops for food, energy, and phytoremediation, through a heavy metal perspective.

Despite the numerous advantages of sewage sludge in the increment of soil chemical characteristics, and the agro-morphological attributes and yields in different crop species, proper screening for heavy metals in all the variants (SS, soil, food products) is a must. Even though most of the results presented in this review were within the legal limits (in the case of soil and SS), crop products presented concerning values for human food consumption in developing countries, which could bring dangerous consequences for human health. In these specific cases, the application of SS and biosolids requires appropriate strict guidelines with appropriate regulatory oversight to control contamination of agricultural soils, there must be a strict control in the application of this bio-resource in the food production and soil amendment. In Europe, this is stated according to the Council Directive 86/278/EEC (on the protection of the environment, and in particular of the soil, when sewage sludge is used in agriculture) [3]; and EU Commission Regulation No. 1881/2006 (by setting maximum levels for certain contaminants in foodstuffs) [41]. It is essential to add value to waste products, of which biosolids are one of the most important ones due to the large amounts of production of this residue in Europe, as already recognized in the FAO Smart Climate Agriculture document [16]. Even regarding the global guidelines for not transcending the boundaries of heavy metals into the soil and food crops, it is important to update and amend the already existing directives, to allow adequate standardization according to several models of application of this residue. If we are facing metal-contaminated ecosystems, the use of energy crops for phytoremediation could be a good sustainable solution for countries that are handling high metal-pollution levels and increasing bioenergy demands. With a previous study of the main ecosystem metal-contaminants and the selection of the best plant phytoremediation strategies to incorporate those heavy metals and metalloids in their tissues, this energy crop-based phytoremediation could be a good mitigation solution with an economic return. However, the harvest-contaminated biomass of the phytoremediation plants should be given special attention due to the ecotoxicological risk for energy production, as also for the possible metal transference contamination with its non-harvested biomass deposition in the ecosystem.

Finally, it is crucial to continue and improve the local implementation of SS and biosolids, either to amend and recover degraded soils or to support the phytoremediation process of metal-contaminated soils, combining the ease of availability of this discarded bioresources and, as a future goal, assessing its potential to promote agrosystems' sustainability. It is fundamental to weigh the pros and cons of using these bioresources, and additional research is needed to allow the study of local food crops' production, bioenergy potential, and amendment needs. Hitherto, one must take into consideration an important risk that is still little addressed in this area: the possible metal leaching after prolonged SS and biosolids applications. Even with its applications under strict legislation and controlled conditions, further studies could play a key role in monitoring and avoiding damage to the surrounding ecosystems.

**Author Contributions:** Conceptualization, N.N.; investigation, N.N., C.R. and C.S.S.G.; writing—original draft preparation, N.N., C.R. and C.S.S.G.; writing—review and editing, N.N., C.R., C.S.S.G. and M.Â.A.P.d.C.; supervision, M.Â.A.P.d.C.; funding acquisition, N.N., C.S.S.G. and M.Â.A.P.d.C. All authors have read and agreed to the published version of the manuscript.

**Funding:** This research was funded by *Programa Operacional Madeira* 14–20, Portugal 2020, and the European Union through the European Regional Development Fund, grant number M1420-01-0145-FEDER-000011 [CASBio].

**Institutional Review Board Statement:** Not applicable.

**Informed Consent Statement:** Not applicable.

**Data Availability Statement:** Not applicable.

**Acknowledgments:** The authors acknowledge the support by National Funds FCT - Portuguese Foundation for Science and Technology, under the projects UIDB/04033/2020 and UIDP/04033/2020.

**Conflicts of Interest:** The authors declare no conflict of interest. The funders had no role in the design of the study; in the collection, analyses, or interpretation of data; in the writing of the manuscript, or in the decision to publish the results.

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
