# Peer review of "Review of Sewage Sludge as a Soil Amendment in Relation to Current International Guidelines: A Heavy Metal Perspective"

_sustainability, doi:10.3390/su13042317_

Round 1

Reviewer 1 Report

This is an interesting review article on a broad topic like sewage sludge, having as basis the legislation and composition analysis of the biosolids, with further ap-plication in soil amendment, crops for food, energy, and phytoremediation, through a heavy metal perspective. The paper’s title match its content. The article have a logical layout. The language of article correct. The paper’s conclusions follow logically from the development of the argument. The text adequately illustrated (tables). 

A few comments for the authors:

  • Line 80: "Food and Agriculture Organi-80 zation (FAO) [15] and the European Directive 86/278/EEC [2] are two main examples of regulations that control..." - FAO is not regulation. The sentence needs to be corrected.
  • Figure 1 should be enlarged (the legend is illegible).
  • Line 233: Who is the author of the pot experiment?

Author Response

Response to Reviewer 1 Comments

Manuscript ID: sustainability-1090195

Type of manuscript: Review

Title: The future of biosolids in agronomy: a heavy metal perspective Authors: Nuno Nunes, Carla Ragonezi *, Carla S. S. Gouveia, Miguel AĚ‚. A. 

This is an interesting review article on a broad topic like sewage sludge, having as basis the legislation and composition analysis of the biosolids, with further application in soil amendment, crops for food, energy, and phytoremediation, through a heavy metal perspective. The paper’s title match its content. The article have a logical layout. The language of article correct. The paper’s conclusions follow logically from the development of the argument. The text adequately illustrated (tables).

The authors appreciate the reviewer comments about the manuscript. The comments were addressed properly below and can be visualized in resubmitted version.

A few comments for the authors: 

Point 1: Line 80: "Food and Agriculture Organization (FAO) [15] and the European Directive 86/278/EEC [2] are two main examples of regulations that control..." - FAO is not regulation. The sentence needs to be corrected.

Response 1: The authors reviewed the information and corrected.

Point 2: Figure 1 should be enlarged (the legend is illegible).

Response 2: The authors corrected the legend´s size according to the reviewer's comment.

Point 3: Line 233: Who is the author of the pot experiment?

Response 3: The authors added the reference.

Reviewer 2 Report

Thank you for submitting your study for publication to Sustainability. This is interesting work and I hope to see it published soon.

Here a few comments that could further improve your work:

-check the style - I feel this is somehow a bit off

-Where is chapter 4

Introduction

-3 step sewage wastewater treatement - include a figure to illustrate that

-15-20% of annual phosphate rock production could be substituted? You cannot substitute reserves - clarfiy

-do the upper levels of heavy metals only apply to SS or also to mineral fertilizers? clarify

-what methods did you use for the review? PRISMA? in any case include a short chapter where you describe them

-since you do not have a traditional article structure shortly explain why you structured your article they way you did

2. Legislation

-include a table were you show and compare the different limits on heavy metal values and average concentrations in SS and mineral fertilizers

-Mineral fertilizers seem to consider considerable amounts of U (Uranium resources in EU phosphate rock imports - ScienceDirect) that I did not see in your analysis - this should make for an interesting comparison

3. Sewage Sludge and ...

-Table1 this is great information, maybe consider a graphical presentation

Author Response

Response to Reviewer 2 Comments

Manuscript ID: sustainability-1090195

Type of manuscript: Review

Title: The future of biosolids in agronomy: a heavy metal perspective Authors: Nuno Nunes, Carla Ragonezi *, Carla S. S. Gouveia, Miguel AĚ‚. A. 

Thank you for submitting your study for publication to Sustainability. This is interesting work and I hope to see it published soon.

The authors appreciate the reviewer comments about the manuscript. The comments were addressed properly below and can be visualized in resubmitted version.

Here a few comments that could further improve your work:

Point 1: check the style - I feel this is somehow a bit off.

Response 1: The authors appreciate the comment and carefully checked the style.

Point 2: Where is chapter 4

Response 2: The authors corrected that information. Please see the manuscript.

Point 3: Introduction

  • 3 step sewage wastewater treatment - include a figure to illustrate that.

Response 3.1: The authors included a figure (fig. 1). Please see the manuscript.

  • 15-20% of annual phosphate rock production could be substituted? You cannot substitute reserves – clarify.

Response 3.2: The information was removed.

  • do the upper levels of heavy metals only apply to SS or also to mineral fertilizers? Clarify.

Response 3.3: The upper values are only for sewage sludge, biosolids and soil when envisioned sludge or biosolids application.

  • what methods did you use for the review? PRISMA? in any case include a short chapter where you describe them.

Response 3.4: The authors included that information as the 2nd chapter in the manuscript.

  • since you do not have a traditional article structure shortly explain why you structured your article they way you did.

Response 3.5: We have designed this structure to individualize the subjects related to the technical benefits and difficulties related to the agronomic application of sewage sludge or biosolids in each theme. Separating the legislation, sewage sludge and biosolids analysis, soil analysis, food crops, phytoremediation, and bioenergetics, we intend to deliver a friendlier review article, that information is easily identified and retrieved. Also, we hope that this work is not only read by researchers but also by policy makers, helping them evaluate and decide the sustainable future of this biomass.

Point 4: Legislation

  • include a table were you show and compare the different limits on heavy metal values and average concentrations in SS and mineral fertilizers.

Response 4.1: The table was added to the manuscript as Table 1.

  • Mineral fertilizers seem to consider considerable amounts of U (Uranium resources in EU phosphate rock imports - ScienceDirect) that I did not see in your analysis - this should make for an interesting comparison.

Response 4.2: The authors appreciate the comment. Sewage sludge contains nitrogen and phosphorous, resulting especially from nitrification-denitrification phases in the wastewater treatment process giving unique fertilizing benefits. Although mineral fertilizers seem to consider considerable amounts of uranium, is difficult to precise how much U is in the SS after treatment, since it will depend strongly on the waste management strategy. This strategy could influence the U content and bioavailability. The topic, while very interesting, and even being a metal, Uranium does not appear in any list of the WHO and FAO guidelines, and in any Directive that was consulted for this manuscript, so the authors prefer not to develop the topic any further in this manuscript.

Point 5: Sewage Sludge and ...

  • Table1 this is great information, maybe consider a graphical presentation.

Response 5: The authors appreciate the comment. Regarding the graphical presentation, authors evaluated the possibility of a graphical presentation of the table but after many trials the authors concluded that was better to present the data as a table.

Reviewer 3 Report

The manuscript "The future of sewage sludge in agronomy: a heavy metal perspective" is having interesting findings. 

My advice, comments and recommendations are listed below:

The language of the manuscript should be more improved so that it is easy to read. You need to correct the grammar. Please go through the entire manuscript and shorten and correct some sentences.

I recommend clarifying and improve presentation of abstract so that it is clear to the reader what this is all about. Include your recommendations and future prospects. It is necessary to extend the abstract with the most significant results.

Please be sure that your manuscript thoroughly establishes how this work is fundamentally novel. Specific comparisons should be made to previously published materials that have a similar purpose. Please present a strong case for how this work is a major advance. This needs to be done in the manuscript itself, not just in the response to review comments.

Please be sure that your abstract and your Conclusions section not only summarize the key findings of your work but also explain the specific ways in which this work fundamentally advances the field relative to prior literature.

The introduction must be expanded and improved. The significance of this study should be more emphasize in the introduction. 

See these paper, you can help you a lot: 

https://www.sciencedirect.com/science/article/abs/pii/S0269749120360632

https://www.sciencedirect.com/science/article/abs/pii/S027312239900373X

Line 45, section heavy metals and wastewaters: This issue has also been addressed in detail in this very important paper, which encourages authors to add it to this place as a reference. https://www.sciencedirect.com/science/article/abs/pii/S0304389420316149

Line 55: ,,Sewage wastewater,, I think it would be appropriate to introduce an abbreviation for example such as SW for this phrase.

Line 92, section adding fly ash and/or lime [19]: Clay minerals or also modified clay minerals with organic cations can also be used for such applications due to their excellent adsorption properties. Please add it to the introduction. This statement support this important paper and therefore it is necessary to add them to this place: https://www.sciencedirect.com/science/article/abs/pii/S0169131719301413

Line 164: Check the correct notation of SI units in the entire manuscript.

Line 187: Improve figure 1, the descriptions are faintly visible.

Line 191: What reason or process do you attribute such a discrepancy in numbers to?

Line 270: Table 1 needs to be significantly improved. In its current form it is confusing. You must also check all the values in this table.

Line 271: The legend under Table 1 should be better explained.

Line 324 - 332: You used a different font size in this section. Fix it.

Line 374: Table 2 needs to be improved, it is confusing. Check all values in the table.

Line 375: Please shorten and simplify the description below Table 2.

Line 444: You need to customize the table so that it is clear and understandable. Check all values in the table.

6. Phytoremediation and Bioenergetics: Write this chapter more clearly. Reformulate some phrases. Make sure that the abbreviations are entered correctly.

7. Conclusions: Indicate the possible risks of such research. Add your recommendations for future research.

Make sure the references are added correctly according to the journal's instructions.

Author Response

Response to Reviewer 3 Comments

Manuscript ID: sustainability-1090195

Type of manuscript: Review

Title: The future of biosolids in agronomy: a heavy metal perspective Authors: Nuno Nunes, Carla Ragonezi *, Carla S. S. Gouveia, Miguel AĚ‚. A.

The manuscript "The future of sewage sludge in agronomy: a heavy metal perspective" is having interesting findings. 

The authors appreciate the reviewer comments about the manuscript. The comments were addressed properly below and can be visualized in resubmitted version.

My advice, comments and recommendations are listed below:

Point 1: The language of the manuscript should be more improved so that it is easy to read. You need to correct the grammar. Please go through the entire manuscript and shorten and correct some sentences.

Response 1: The authors reviewed and corrected when necessary.

Point 2: I recommend clarifying and improve presentation of abstract so that it is clear to the reader what this is all about. Include your recommendations and future prospects. It is necessary to extend the abstract with the most significant results.

Response 2: The authors appreciate the comment. Due to the word limitation, we could not extend the abstract any further, but recommendations and prospects were included. Please see in the manuscript.

Point 3: Please be sure that your manuscript thoroughly establishes how this work is fundamentally novel. Specific comparisons should be made to previously published materials that have a similar purpose. Please present a strong case for how this work is a major advance. This needs to be done in the manuscript itself, not just in the response to review comments.

Response 3: The authors introduced the information according to the review´s comment. Please see mainly in the introduction.

Point 4: Please be sure that your abstract and your Conclusions section not only summarize the key findings of your work but also explain the specific ways in which this work fundamentally advances the field relative to prior literature.

Response 4: The authors introduced the information according to the review´s comment. Please see in the manuscript.

Point 5: The introduction must be expanded and improved. The significance of this study should be more emphasize in the introduction. See these paper, you can help you a lot:

https://www.sciencedirect.com/science/article/abs/pii/S0269749120360632

https://www.sciencedirect.com/science/article/abs/pii/S027312239900373X

Response 5: The authors appreciate the comment and improved the introduction. Additionally, the recommended bibliography was incorporated in the manuscript.

Point 6: Line 45, section heavy metals and wastewaters: This issue has also been addressed in detail in this very important paper, which encourages authors to add it to this place as a reference. https://www.sciencedirect.com/science/article/abs/pii/S0304389420316149

Response 6: The authors appreciate the comment, and the recommended bibliography was incorporated in the manuscript.

Point 7: Line 55: ,,Sewage wastewater,, I think it would be appropriate to introduce an abbreviation for example such as SW for this phrase.

Response 7: The abbreviation was added as recommended.

Point 8: Line 92, section adding fly ash and/or lime [19]: Clay minerals or also modified clay minerals with organic cations can also be used for such applications due to their excellent adsorption properties. Please add it to the introduction. This statement support this important paper and therefore it is necessary to add them to this place: https://www.sciencedirect.com/science/article/abs/pii/S0169131719301413

Response 8: The authors appreciate the comment, and the recommended bibliography was incorporated in the manuscript.

Point 9: Line 164: Check the correct notation of SI units in the entire manuscript.

Response 9: The authors checked and corrected when necessary.

Point 10: Line 187: Improve figure 1, the descriptions are faintly visible.

Response 10: The authors corrected the legend´s size according to the proceeds to the reviewer's comment.

Point 11: Line 191: What reason or process do you attribute such a discrepancy in numbers to?

Response 11: The authors added to the text one of the reasons for the discrepancy.

Point 12: Line 270: Table 1 needs to be significantly improved. In its current form it is confusing. You must also check all the values in this table.

Response 12: The authors appreciate the comment. We compiled the information in this table to simplify the amount of information that was gathered from the articles. In the left column you have the locations and in the top line the elements. The values were checked.

Point 13: Line 271: The legend under Table 1 should be better explained.

Response 13:  The authors clarify the information.

Point 14: Line 324 - 332: You used a different font size in this section. Fix it.

Response 14: The authors corrected the font size.

Point 15: Line 374: Table 2 needs to be improved, it is confusing. Check all values in the table.

Response 15: The authors appreciate the comment. We compiled the information in this table to simplify the amount of information that was gathered from the articles. In the left column you have the locations and in the top line the elements. The values were checked.

Point 16: Line 375: Please shorten and simplify the description below Table 2.

Response 16: The authors simplified the description of table 2.

Point 17: Line 444: You need to customize the table so that it is clear and understandable. Check all values in the table.

Response 17: The authors appreciate the comment. We compiled the information in this table to simplify the amount of information that was gathered from the articles. In the left column you have the specie/plant organ and in the top line the elements and the SS application and locations. The values were checked.

Point 18: 6. Phytoremediation and Bioenergetics: Write this chapter more clearly. Reformulate some phrases. Make sure that the abbreviations are entered correctly.

Response 18: We appreciate the reviewer’s comment. To better clarify this section, we rearranged some of the sentences, and we tried to improve the text connection.

Point 19: 7. Conclusions: Indicate the possible risks of such research. Add your recommendations for future research.

Response 19: The suggestions regarding the conclusions were taken into consideration. Beyond main inferences on our research in the sewage sludge applications and care for soil amendment, crops for food, energy, and phytoremediation, we complement the conclusion content with its potentiality to promote agrosystems sustainability, recommending the need for additional research to allow the study of local food crops production, bioenergy potential, and amendment needs. Hitherto, we also highlight one potential risk that is still little addressed in this area: the possible metal leaching after prolonged SS applications, which should be monitored to avoid damage to the surrounding ecosystems.

Point 20: Make sure the references are added correctly according to the journal's instructions.

Response 20: The authors checked and corrected when necessary.

Round 2

Reviewer 1 Report

The manuscript after the changes is definitely better.

Reviewer 2 Report

Good work

Reviewer 3 Report

This manuscript has been significantly improved.